# Difficulty Evaluation of Navigation Scenarios for the Development of Ship Remote Operators Training Simulator

Taemin Hwang [1] and Ik-Hyun Youn [2,*]

1   Department of Maritime Transportation System, Mokpo National Maritime University, Mokpo 58628, Korea
2   Division of Navigation and Information Systems, Mokpo National Maritime University, Mokpo 58628, Korea
*   Correspondence: iyoun@mmu.ac.kr

**Abstract:** The enhancement of navigators' ability has been promoted by on-scene training; however, considering the safety and repeatability, simulation training (ST) is recommended. Notably, the training of maritime autonomous surface ship (MASS) remote operators has to be performed in a systemic simulated environment. In various fields, ST has differentiated levels of training scenarios considering the proper training effect and evaluation. Although the accuracy and implementation of a realistic situation have received the most attention in simulated navigation, the objective criteria of difficulty are to be established for systemic training. For this purpose, this study aims to propose difficulty criteria in navigation generation scenarios for the development of training simulator MASS remote operators. Proposed methods generated navigation scenarios with differentiated difficulties, simulated navigation experiments were performed, and the results were analyzed as a validation of the differentiated difficulties. Our findings include the difficulty differentiation method, navigation scenario samples, and simulated navigation experimental results.

**Keywords:** simulated training; remote operator; navigation scenario generation; differentiated difficulty; simulated navigation experiments

## 1. Introduction

In Maritime Education and Training (MET), enhancing ship operators' maneuvering ability has been the primary purpose of training [1,2] because it is strongly related to maritime accidents [3]. At present, on-scene training is obligatory for trainees [4,5], although opportunities to maneuver a ship as a cadet in situations that require attention are somewhat limited for rational reasons for safety [6]. Instead, simulation training (ST) is performed for safety and repeatability [7–9]. As the Maritime Autonomous Surface Ship (MASS) is in development [10,11], requiring remote operators' intervention in levels of autonomy 2 and 3 [12,13], the required navigation proficiency for the remote operators will be considerable [14–16]. Before engaging in actual remote operation in the shore control center, the training of remote operators is encouraged to be performed in a systemically simulated training environment [9,17], the shore remote control simulator.

In the development of a systemic shore remote control simulator, the various evaluation methods in other fields using ST were researched. Aviation training used a scoring system for trainees' actions, providing evaluation results in scenarios under different designated difficulties to allow trainees to understand their weak points [18]. Likewise, data-driven methods are used to consider the reaction time and delay in evaluating a pilot's flight abilities [19]. In medical training, patients in different levels of situations are given in simulation, and actions taken by trainees are evaluated for each level [20], or the simulated surgical suturing performance results of trainees are evaluated using data-driven methods [21]. Even in safety management training, ST results are evaluated in different situations using a checklist [22]. After the simulation, training is performed on different levels of situations for the systemic evaluation of trainees.

However, in the maritime field, simulation-related research has mainly focused on the fidelity of simulation [23–26], the implementation of ship motions, and real situations [27–31]. Fidelity and reality are essential factors in navigation training, but difficulty evaluation is lacking in systemic ST. Regarding the difficulty in ST, the adjustment of navigation elements' complexity and the composition of navigation elements are under the control of instructors [26,32] so that navigation scenarios can be either easy or difficult. This research concentrated on establishing objectively differentiated difficulty degrees, particularly when generating navigation scenarios considering repeatability and reproductivity. To differentiate the difficulty of navigation scenarios objectively, research methods derived the actual distribution of navigation elements, generated navigation scenarios, performed a navigation experiment, and validated the results.

Therefore, the study proposed objective methods to differentiate navigation difficulties for the development of a systemic training simulator for MASS remote operators.

## 2. Materials and Methods

The proposed methods comprise four steps separated into two sections. In the first step, the extraction of navigation elements and distribution fittings is based on a previous study [17]. Then, the difficulty degrees were set to generate navigation scenarios. Afterward, a simulated navigation experiment was performed, features were engineered, and the difficulty of the navigation scenarios was validated. Figure 1 shows the workflow of the proposed methods.

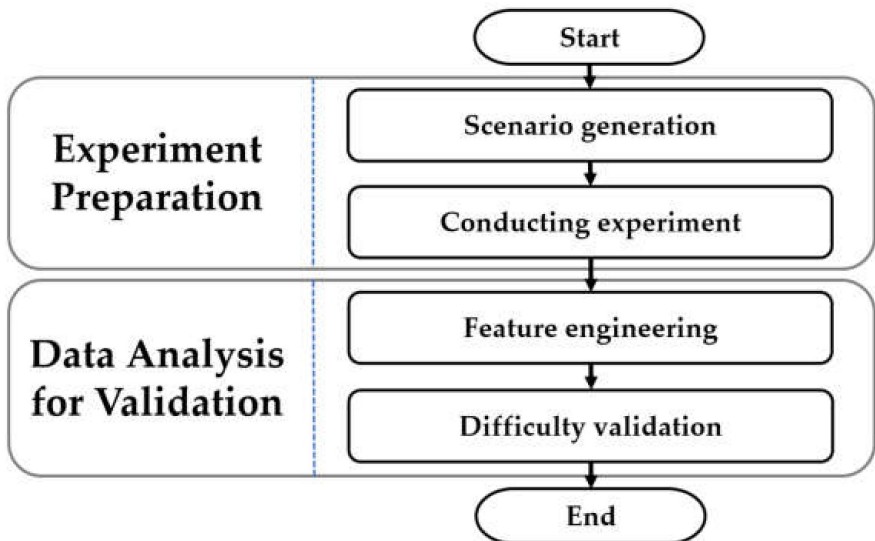

**Figure 1.** Workflow of proposed methods.

The "Experiment Preparation" section includes the navigation scenario generation for the simulated navigation experiment, and the "Data Analysis for Validation" section includes the maneuvering feature engineering for difficulty clustering.

### 2.1. Navigation Scenario Generation

2.1.1. Data Collection and Preprocessing

The AIS trajectory data were collected for two months: January and August 2020. The data collection conditions were described. First, gross tonnage conditions targeted ships that are neither too small nor too large for an autonomous ship of the development project in South Korea. Second, speed conditions excluded anchored and drifting ships. Third, the covered area was limited to the Busan port entrance, where ships made dynamic movements. The specific data collection conditions are summarized in Table 1.



**Table 1.** Data collection condition.

| Criteria | Condition Range | Unit |
|---|---|---|
| (1) Gross tonnage | 80,000–120,000 | - |
| (2) Speed | 5–25 | Knots |
| (3) Covered area (Latitude) | 34.85–35.97 | Degrees |
| (4) Covered area (Longitude) | 128.85–129.00 | Degrees |

### 2.1.2. Difficulty Degrees of Navigation Elements

Considering our experiment, this research extracted only two navigation elements: the "course-altering angle" and "straight proceeding distance". When extracting navigation elements, a 5-min window and five threshold degrees were used to determine whether a ship changed course. After the extraction, the ranges of navigation elements were reduced to a suitable size for our experiment scenarios: a maximum of 60° angle and one nautical mile of distance. Figure 2 shows the navigation elements fitted in probability distribution curves and difficulty degrees.

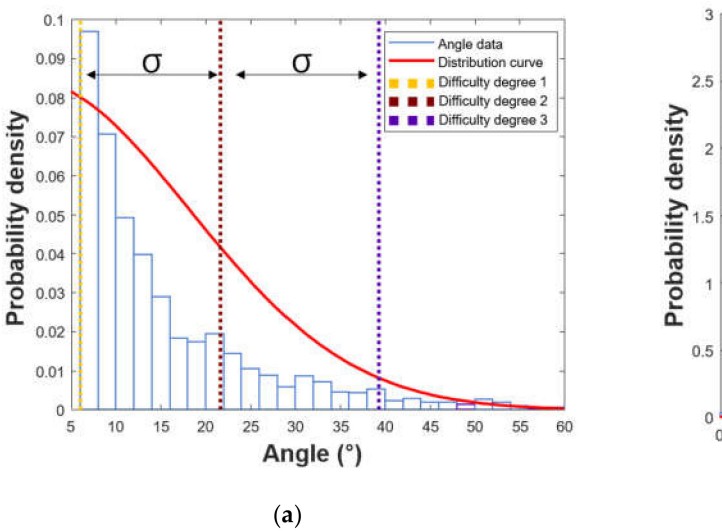
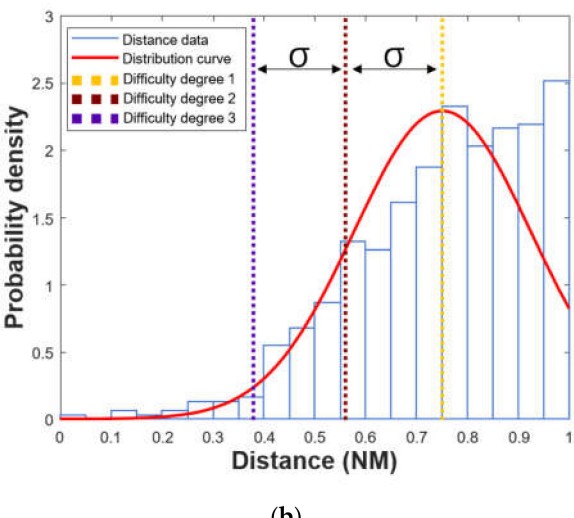

(**a**)          (**b**)

**Figure 2.** Distribution of navigation elements and difficulty degrees: (**a**) "Course-altering angle" with a higher degree for a larger value and (**b**) "Straight proceeding distance" with a higher degree for a smaller value.

Setting difficulty degrees requires the "minimum difficulty degree" and "difficulty gap." Starting from the mean value as Difficulty degree 1, the sigma value was added or deducted to set Difficulty degrees 2 and 3. Table 2 shows the difficulty gaps and degrees.

**Table 2.** Difficulty degrees of navigation elements.

| Navigation Elements | Gap (Sigma Value) | Degree 1 | Degree 2 | Degree 3 |
|---|---|---|---|---|
| Course-altering angle (°) | 17 ($\sigma$) | 5 | 22 (5 + $\sigma$) | 39 (5 + 2$\sigma$) |
| Straight proceeding distance (NM) | 0.17 ($\sigma$) | 0.72 ($\mu$) | 0.55 ($\mu - \sigma$) | 0.38 ($\mu - 2\sigma$) |

For the "course-altering angle," the mean value was nearly zero due to the symmetry shape of the distribution (portside and starboard side). Instead, this research used a 5° angle as the minimum difficulty degree.

### 2.1.3. Navigation Scenarios in Differentiated Difficulties

Regarding the proposed concept of difficulty, the low-difficulty scenario has a more minor course-altering angle and a longer straight proceeding distance to the next waypoint,

giving trainees sufficient time and place for ship maneuvering along the navigation route. Unlike the low-difficulty, the high-difficulty scenario has a larger course-altering angle and shorter straight proceeding distance to the next waypoint, making the trainees feel challenged to navigate. To determine such difficulty, this research borrowed the "difficulty and importance matrix" concept. Figure 3 shows our proposed difficulty matrix, and Figure 4 shows navigation scenarios in differentiated difficulties for our experiments.

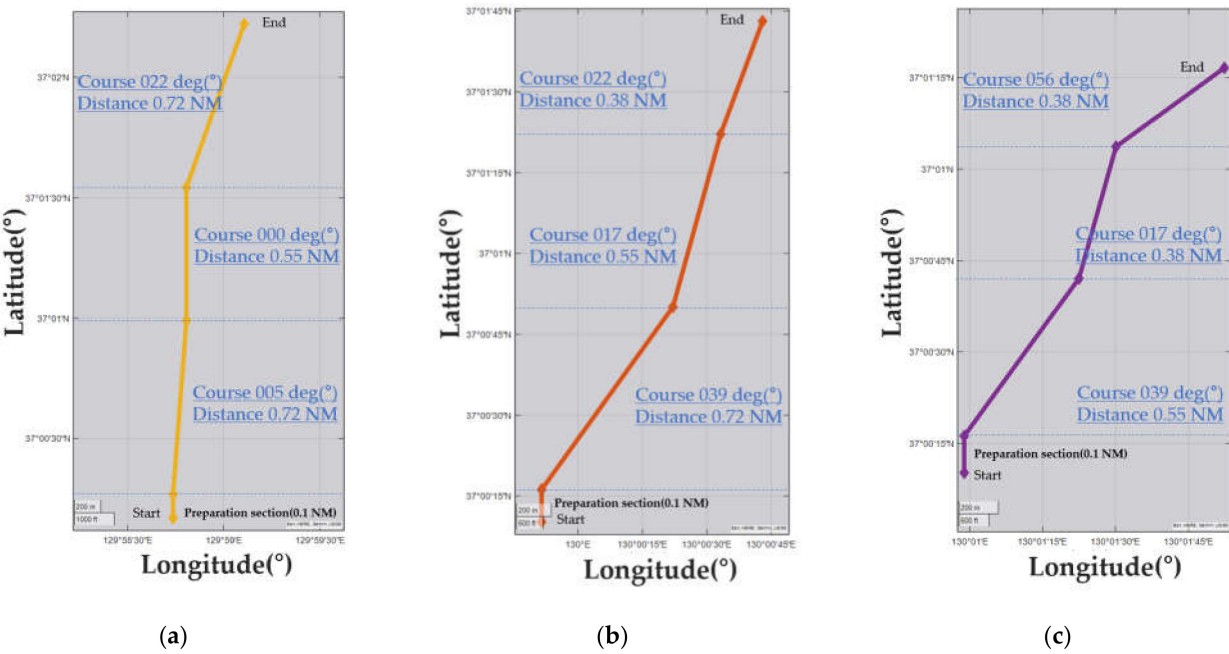

**Figure 3.** Difficulty matrix for navigation scenario generation.

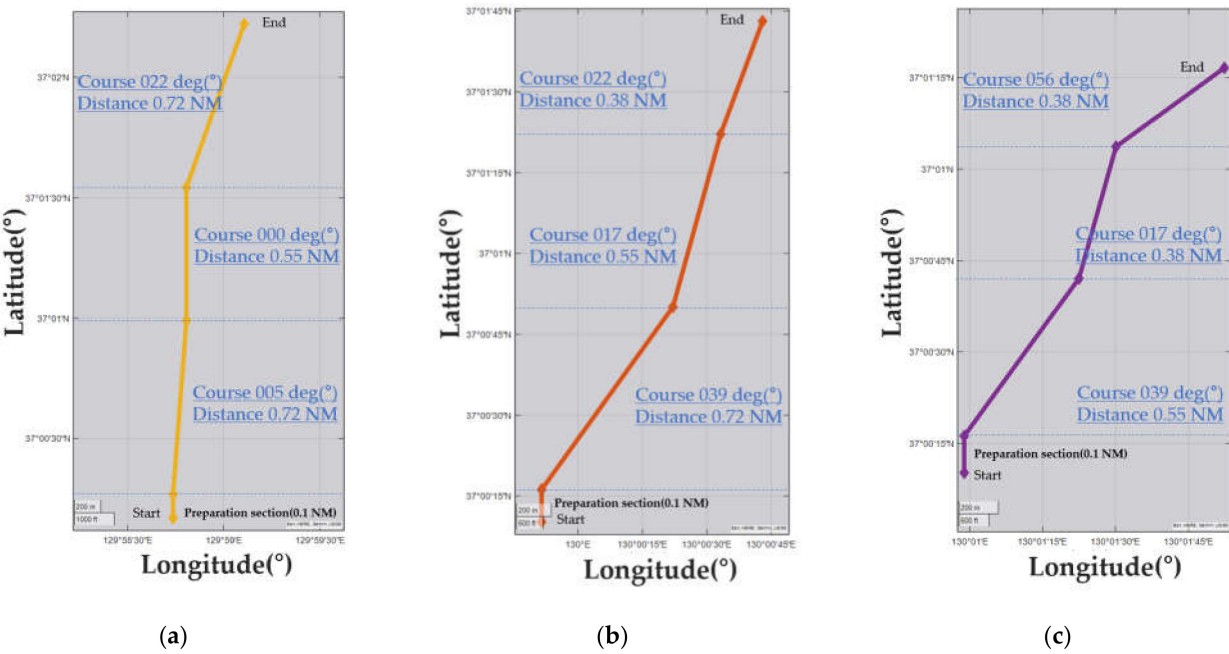

**Figure 4.** Navigation scenarios in differentiated difficulties: (**a**) low, (**b**) medium, and (**c**) high difficulties.

The above navigation scenarios depict our difficulty concept for two navigation elements. A smaller course alteration and more prolonged straight proceeding have less difficulty.

### 2.2. Navigation Experiment Execution

2.2.1. Experiment Configuration

This research performed simulated navigation experiments at Mokpo National Maritime University. Then, data were collected using a full-mission-ship-handling simulator.

All participants have experience in onboard training, thereby possessing the basic knowledge of handling navigation equipment [33]. Specific characteristics of participants are listed in Table 3.

**Table 3.** Participants' characteristics.

| Characteristics | Description |
|---|---|
| (1) Number of participants | 34 |
| (2) Grade | Senior grade student |
| (3) Onboard training period | 11–12 months |
| (4) Type of ship in onboard training | Merchant and training ships |

2.2.2. Experiment Protocol

The simulated navigation was performed for 45 min for each participant, the ship proceeded at 13-knot speed in an open sea environment without any obstacle, and the navigation route was charted on an Electronic Chart Display Information System. Although all participants completed a year-long onboard navigation training, the specific use of navigation equipment, as well as the navigation scenarios in differentiated difficulties, was demonstrated to each participant. Figure 5 shows participants performing simulated navigation.

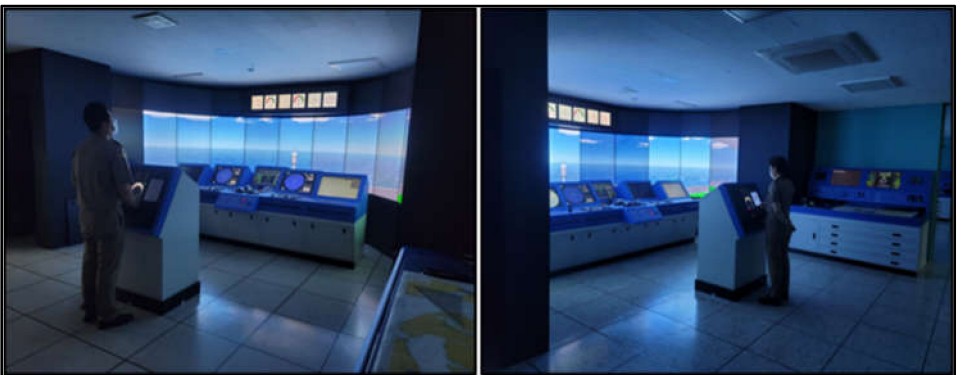

**Figure 5.** Images of participants in a simulated navigation environment.

*2.3. Maneuvering Feature Extraction*

The data was collected including "time", "location", and "rudder angle", from the simulator to extract maneuvering features (Table 4). In extracting features, navigation experience and the experimental protocol were considered.

**Table 4.** Maneuvering features.

| Domain | Abbreviation | Feature Specification | Unit |
|---|---|---|---|
| Steering | ART | Average rate of turn | °/min |
| | SRT | Standard deviation of the rate of turn | °/min |
| | ARD | Average rudder angle | ° |
| Spatial | ODD | Overall distance difference | % |
| | AXD | Average cross-track distance | cable |
| | SXD | Standard deviation of cross-track distance | cable |
| Temporal | OTD | Overall time difference | % |
| | ORM | Overall time of rudder angle at "0" | % |

*2.4. Difficulty Validation*

Among maneuvering features derived in the previous subsection, ineffective features were eliminated using a stepwise regression method. After eliminating features whose

*p*-value is more significant than 0.05 based on the sum of square error (SSE) criterion, this study performed t-distributed stochastic neighbor embedding (t-SNE) clustering to validate if the proposed differentiated difficulties are adequately divided into clusters.

## 3. Results

After our simulation experiments using navigation scenarios in differentiated difficulties of our proposed methods, the experimental results were analyzed to validate differentiated difficulties. In this section, the research focused on feature engineering and validation, as navigation scenario generation was described in the previous section.

### 3.1. Simulation Experimental Results

Because this research presupposed that the goal of the experiment is perfect ship maneuvering, participants struggled to maintain the ship's course and distance on the charted route. Some participants exhibited superb maneuvering, whereas others struggled. Figure 6 shows the ship's trajectory for three difficulties of navigation scenarios.

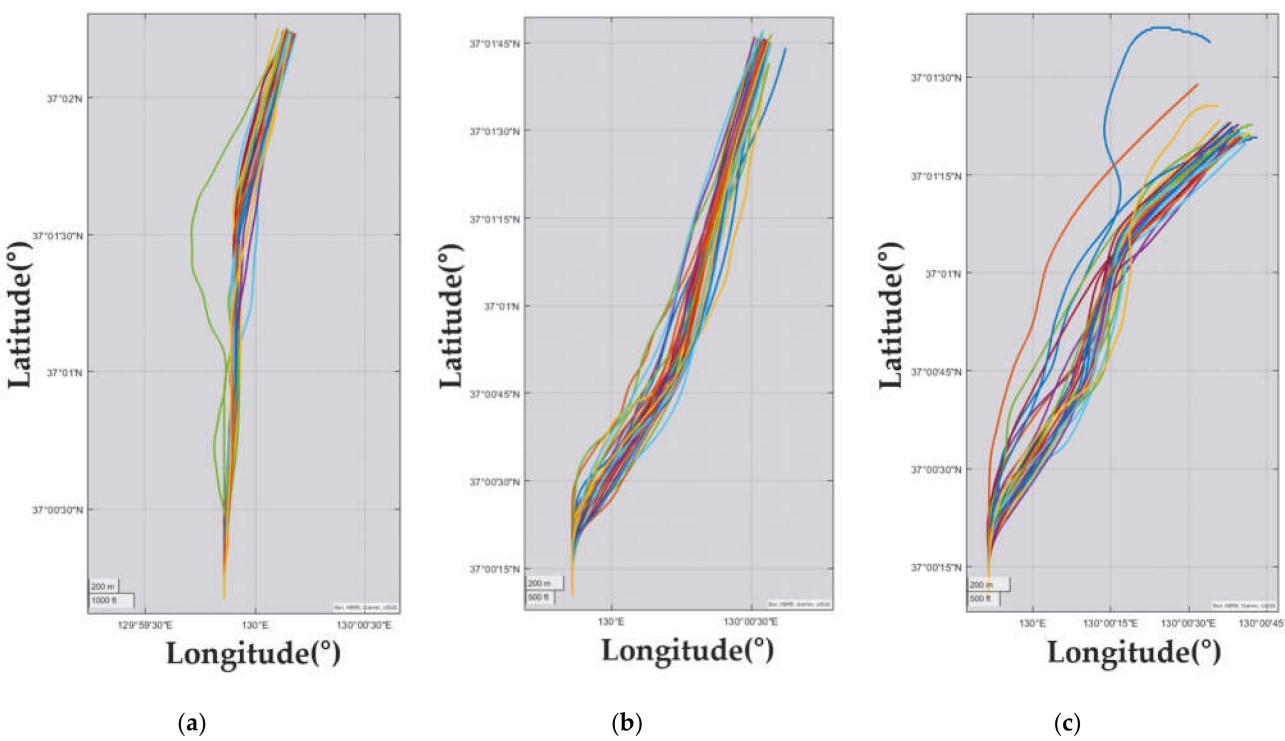

(**a**)          (**b**)          (**c**)

**Figure 6.** Ship's trajectory data collected from experiments: (**a**) low, (**b**) medium, and (**c**) high difficulties.

In the experiments, engine controlling was not allowed to set the experiment condition equally and to prevent unexpected maneuvering from disturbing the navigation goal—perfect maneuvering along the charted route.

### 3.2. Feature Extraction

Maneuvering features (Table 4) are in divided performance domains where different navigation factors were considered. The steering domain includes features related to "rudder" and "ship's turning rate", which decide whether the ship's movement is rough or consistent. Similarly, the spatial domain has features regarding over proceeded distance, and the temporal domain has exceeded or elapsed time as features. To compare features objectively, z-score normalization was applied. Figure 7 depicts the maneuvering features in the divided performance domains.

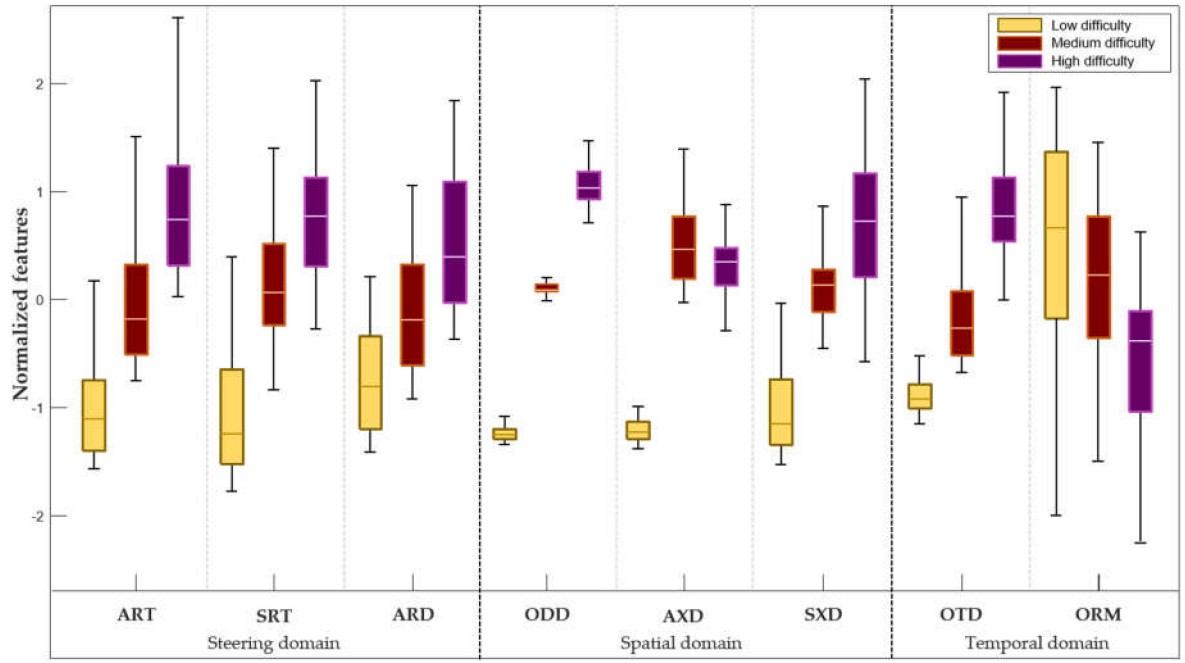

**Figure 7.** Maneuvering features and domains.

### 3.3. Scenario Difficulty Validation

#### 3.3.1. Feature Selection

Considering the relevance of domains in actual navigation performance, an examination was conducted on each participant's variance of navigation along the navigation scenarios at differentiated difficulties. From the perspective of altering course and maintaining course, the rudder use gets rougher as difficulty increases. Figure 8 shows the rudder use and trajectory data of the first participant (Participant 1) as a sample.

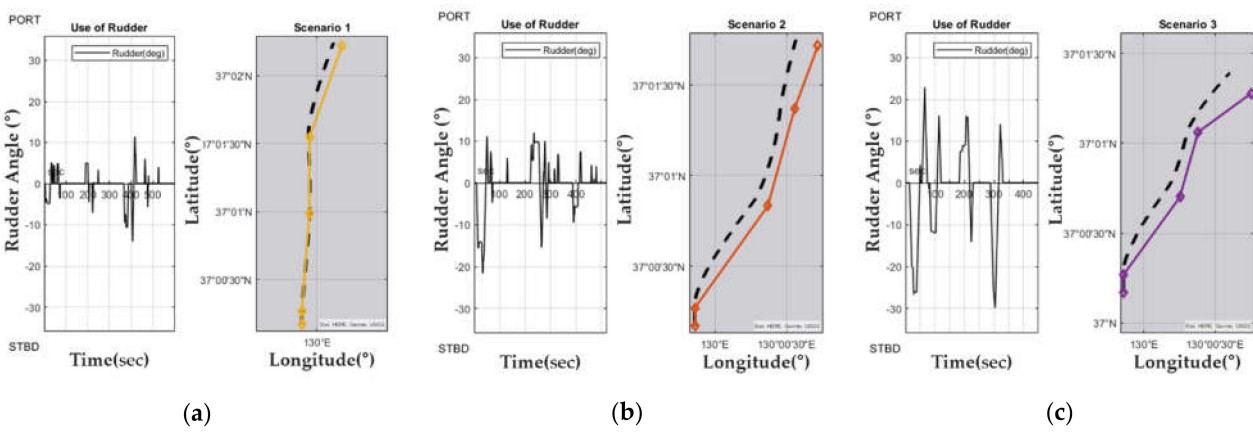

(a)            (b)            (c)

**Figure 8.** Participant 1's rudder and ship trajectory results: (**a**) low, (**b**) medium, and (**c**) high difficulties.

Among the aforementioned maneuvering features, we selected five using the stepwise regression method based on the SSE minimum: ARD, ODD, SXD, OTD, and ORM. Considering that features in the steering domain are similar, the stepwise regression method derived reasonable results. Therefore, these features were selected and applied to the t-SNE clustering algorithm.

#### 3.3.2. Difficulty Validation

To validate how the differentiated difficulty affected navigation performance, this research first combined selected features into X and Y using the t-SNE clustering algorithm

and then compared labeled combined features with k-means clusters. In addition, the lines are connected; subject 1 was described with dots, arrows, and gray colored circles as a representative sample, and additional data was shown to track each participant's results along the differentiated difficulties in Figure 9.

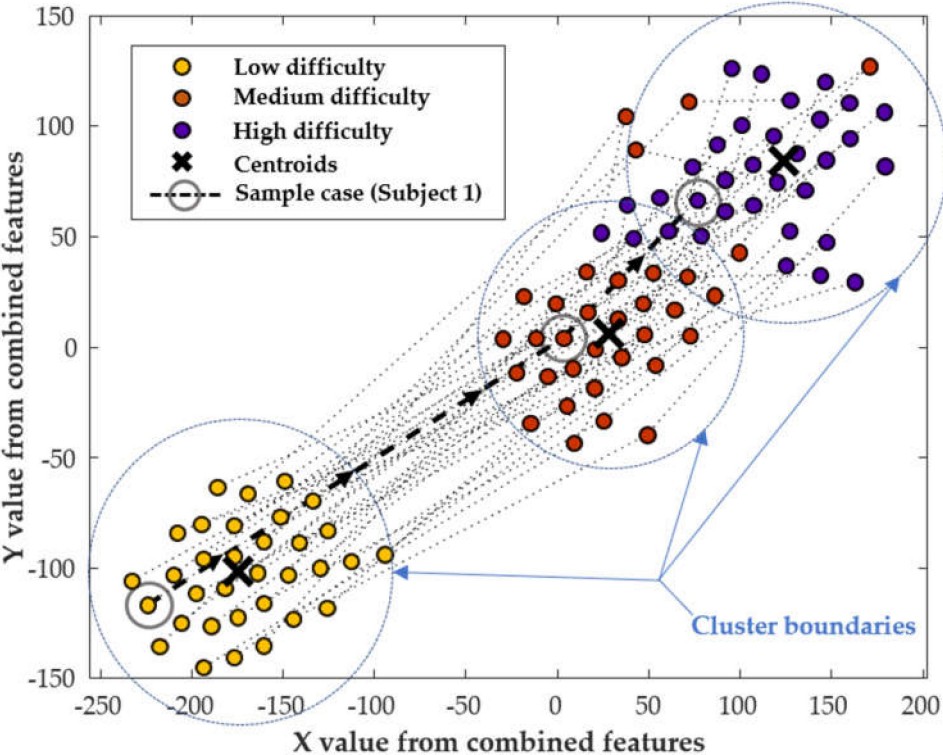

**Figure 9.** Separated clusters along differentiated difficulties.

## 4. Discussion

This research proposed a navigation scenario generation model considering the differentiated difficulties. Our findings are discussed below.

### 4.1. Simulation Experiment

When controlling MASS remotely, the navigation situation given to the operator may not allow course changing, or them to request immediate course changing. In preparing for those situations, trainees should be used to the ship's movement. The ship's trajectory results in Figure 6 depict participants' decisions according to the differentiated difficulties. On the one hand, the trajectories of the low difficulty have a relatively narrow width, indicating trajectories concentrated in the center with only a few trajectories off the route. On the other hand, in the medium and high difficulties, the width of trajectories got larger and weirder, losing their shapes. Clearly, the participants felt that handling the ship became more challenging with a smaller room to steer and a tighter time to make decisions. In the given one cable of preparation section, the width of trajectories began to differ between participants who chose to alter course earlier due to the ship's late movement feedback and those who could not consider this. This is because handling a ship requires understanding the ship's movement according to its speed, size, and other information. However, the data and equipment used in the experiments were from conventional ships, so when the MASS remote operator training simulator is developed, it is necessary to consider the difference between the features of conventional ships and MASS.

### 4.2. Features in Domains

Even though the controlling of MASS could use other types of steering, still the ship maneuvering requires operators to understand how to steer in consideration of the ship's

present location and expected arrival time to the following location. Table 4 and Figure 7 have eight features in three domains, and each domain has a different aspect of ship maneuvering. The steering domain has three features, which indicate only how proficiently the test participants steered in the charted route. Therefore, Features could be helpful when it is necessary to assess the course-maintaining skill, even if the test participants did not strive to follow the charted route. Meanwhile, the spatial domain features exhibit how close the trajectory to the charted route was. No matter what rudder angle a test participant used or how rapidly the ship altered, these features only tell the distance difference. Likewise, the temporal domain features indicate the time difference or accumulated time of action solely. Accordingly, features were selected evenly from each domain to evaluate the full maneuvering results of our experiments.

*4.3. Scenario Difficulty Validation*

4.3.1. Feature Selection with Interpretation

Considering features are based on the conventional ship simulation experiment, not the MASS simulation yet, it was preferred to use most of the maneuvering domains evenly as possible. In the sample test results in Figure 8, Participant 1 steered relatively little and kept the gap with the charted route small in the low-difficulty scenario. The steering was not conspicuously rougher in the medium-difficulty scenario, but the gap with the charted route was significantly greater than that in the low-difficulty scenario. Contrariwise, in the high-difficulty scenario, the steering was conspicuously rougher than in the medium-difficulty scenario when the gap with the charted route was not outstanding. In addition, features in the maneuvering domains worked differently when distinguishing difficulties. Fortunately, the stepwise regression model results were under our expectations. Five features: ARD, ODD, SXD, OTD, and ORM, were selected. Those features may differ when test experiments are conducted in the MASS remote operator training simulators, but presently, it was the appropriate result. The features are selected evenly from domains leaving the matter of controller types flexible in future research.

4.3.2. Difficulty Validation with Interpretation

Combined features in clusters are displayed using the t-SNE clustering and k-means algorithms in Figure 9. As reasons for using these algorithms, the t-SNE clustering algorithm is well-suited for visualizing high-dimensional features and keeping their effect the same, and the k-means algorithm is appropriate for measuring centroids for distance calculation among clusters. In Figure 9, Participant 1 as a sample shows differentiated results from low to high difficulties; other participants show similar results. Most participants were close to the centroid of clusters, but some were outside the cluster boundaries. The participants out of boundaries were found to conduct the previous navigation scenario badly and concentrated more on the following navigation scenario to compensate for mistakes. This can happen in the actual navigation, but a second chance will not be given. In terms of distances among centroids, the low and medium difficulties were derived more than 1.5 times longer than the medium and high difficulties. Consequently, the results showed well-separated differentiated difficulties, but the adjustment of the difficulty criterion was found to be necessary.

**5. Conclusions**

This study proposed difficulty differentiation methods for simulated navigation scenarios that can be used in the training of MASS remote operators. Research methods include the actual navigation element distribution extraction, navigation scenario difficulty division, simulation experiment, and difficulty evaluation using a clustering algorithm. Those were not involved in other related research since they adjusted complexity navigation elements only to differ the navigation difficulties. Our findings indicate that the steps of navigation difficulty can differ objectively. In the future, the study expects remote operators to take charge of MASSs' safe navigation, which requires genuine navigation proficiency

and situations recognizing senses from repeated training. In this regard, using proposed methods to generate massive navigation scenarios under systemically evaluated navigation difficulty can support the development of a training simulator for MASS remote operators. The authors would develop an advanced model with more navigation elements for the implementation of the MASS remote operator training simulator without the limitations found in this research.

**Author Contributions:** Conceptualization, T.H. and I.-H.Y.; Data curation, I.-H.Y.; Formal analysis, T.H.; Funding acquisition, I.-H.Y.; Methodology, T.H.; Project administration, I.-H.Y.; Supervision, I.-H.Y.; Visualization, T.H.; Writing—original draft, T.H.; Writing—review & editing, I.-H.Y. All authors have read and agreed to the published version of the manuscript.

**Funding:** This research and APC were funded by the Ministry of Oceans and Fisheries, Korea in grant number 202106314.

**Institutional Review Board Statement:** The internal institutional review board determined to exempt the ethical review and approval for this study due to the ship handling simulator used in the experiment being a simulator used for education and training generally on the maritime institute. Also, the experiment was conducted in an environment without any physical restrictions on the subject's body.

**Informed Consent Statement:** Not applicable.

**Data Availability Statement:** Not applicable.

**Acknowledgments:** This research was a part of a project titled "Development of Smart Port-Autonomous Ships Linkage Technology", supported by the Ministry of Oceans and Fisheries, Korea.

**Conflicts of Interest:** The authors declare no conflict of interest.

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
