# Peer review of "Difficulty Evaluation of Navigation Scenarios for the Development of Ship Remote Operators Training Simulator"

_sustainability, doi:10.3390/su141811517_

Round 1

Reviewer 1 Report

Kindly refer to attachment.

Author Response

Dear, reviewer

Thank you so much for the precious comments. The authors revised the manuscript and arranged an attached cover letter.

Kindly request to review the revised version of the manuscript again.

Thank you.

Reviewer 2 Report

The authors performed difficulty criteria in navigation generation scenarios for training MASS remote operators, the manuscript touches upon relevant and valid topics. In this proposed framework, the navigation scenarios with differentiated difficulties were generated, simulated navigation experiments were performed, and the results as a validation of the differentiated difficulties were analyzed. This work seems that contributes to an interesting theme, and is properly structured and rooted in the recent literature for the given purpose. The value of this manuscript should not be ignored. However, there is a large gap between content and title.

1. The theme of this manuscript is focused on MASS. At first, I read this article with strong interest, but unfortunately, I didn't get the results that I wanted to see until the conclusion. The all used data is based on conventional ships. The control type and operation features of MASS have not been considered. The structure and function of the shore control center also are not specified. The training scenarios should be different in levels of autonomy, etc.

None of these questions have been answered, which is the deadliest problem of this manuscript. The training of MASS remote operators is the possible application of this manuscript, but it is not specifically designed for MASS.

2. From the introduction, the authors only claimed the application of ST in the maritime field, the potential application or benefits for MASS have not been involved. Especially, There is a serious shortage of cited papers on autonomous ships. Some important and high-value papers have not been cited. If the authors want to establish the link with autonomous ships, the latest reference of autonomous ships needs to be referred. Meanwhile, some features of autonomous ships must be considered, not just the LOA of ship. The phase of operation of MASS needs to be defined, such as L2? L3? L4?

3. In my view, the paper is more like an interesting generation of difficulty criteria in navigation simulator. The conclusion of difficulty criteria is also obvious, which is not focused on MASS. The title needs to be modified to eliminate unnecessary misunderstandings.

4. In line 84, 1 mile should be confirmed. 1 mile? Or 1 n mile?

5. The discussion is based on the achieved results to a very limited extent. A clear link between the results and their analysis must be established.

6. The discussion or conclusions shall discuss the potential uncertainties of the study and its limitations.

7. Careful proofreading is required, as there exist typos and expressive mistakes.

Author Response

(The authors gave the same response as above.)

Round 2

Reviewer 1 Report

It is accepted for publication.

Author Response

Dear reviewer, 

Thanks to your great comment, this paper's writing got way better. 

Thank you so much, and have a great day

Reviewer 2 Report

1. The potential uncertainties of the study and its limitations should be discussed in Section Discussion, instead of Conclusion.

2. Line 29, the expression "autonomous levels" is unprofessional, it should be "level of autonomy".

3. The number of references related to MASS is few, and the newest  MASS reference should be considered to add.

Based on the above, I recommend that the manuscript be accepted not earlier than once the above issues are properly addressed.

Author Response

Dear, reviewer. 

Thank you for the quick response and the comments on the revised manuscript. The authors revised the relevant parts and marked them in magenta. Please see the attachment for the details.

Thank you so much
